🔓 | **Open Peer Review** | Pathogenesis and Host Response | Research Article

# Probiotic *Enterococcus faecalis* surface-delivering key domain of EtMIC3  proteins: immunoprotective efficacies against *Eimeria tenella* infection in chickens

Xinghui Pan,[1] Rui Kong,[1] Qiuju Liu,[1] Zhipeng Jia,[1] Bingrong Bai,[1] Hang Chen,[1] Wenjing Zhi,[1] Biao Wang,[1] Chunli Ma,[2] Dexing Ma[1,3]

**ABSTRACT**  *Enterococcus faecalis* MDXEF-1 strain was used as host bacteria to deliver surface-displayed key domains BC1 and C4D of the *Eimeria tenella* microneme 3 (EtMIC3) protein, BC1 fusion to dendritic cell (DC) targeting peptides (DCpep) (DC-BC1) and DC-C4D, respectively. The expression of target proteins in MDXEF-1/BC1-CWA, MDXEF-1/DC-BC1-CWA, MDXEF-1/C4D-CWA, and MDXEF-1/DC-C4D-CWA was verified. The systemic immune responses induced by oral immunization with recombinant *E. faecalis* were recorded. The immunized chickens were challenged with $4.0 \times 10^4$ *E. tenella*-sporulated oocysts. The immune protections were evaluated by calculating average weight gain and oocyst reduction rate, observing gross and histopathological changes in the cecal tissues, and quantifying levels of inflammatory cytokines in the cecal tissues. The results showed that proteins BC1, C4D, DC-BC1, and DC-C4D were expressed on the surface of *E. faecalis*, respectively. Oral immunizations with recombinant *E. faecalis*, especially MDXEF-1/DC-BC1-CWA and MDXEF-1/DC-C4D-CWA, stimulated higher levels of antigen-specific humoral and intestinal mucosal immune responses, higher levels of Th1, Th2, and Th17-type cytokines, and higher proliferation of peripheral blood lymphocytes than phosphate-buffered saline (PBS) and vector control groups ($P < 0.01$). The groups immunized with four recombinant *E. faecalis* showed better protective effects against homologous infection than the vector control and infection control groups, and the MDXEF-1/DC-BC1-CWA group displayed the best effects. These results demonstrated that probiotic *E. faecalis* delivering DCpep fused with the key domain of the EtMIC3 protein could be a potential approach for the prevention of *Eimeria* infection.

**IMPORTANCE**  Avian coccidiosis caused by *Eimeria* brings huge economic losses to the poultry industry. Although live vaccines and anti-coccidial drugs were used for a long time, *Eimeria* infection in chicken farms all over the world commonly occurred. The exploration of novel, effective vaccines has become a research hotspot. *Eimeria* parasites have complex life cycles, and effective antigens are particularly critical to developing anti-coccidial vaccines. Microneme proteins (MICs), secreted from microneme organelles located at the parasite apex, are considered immunodominant antigens. *Eimeria tenella* microneme 3 (EtMIC3) contains four conserved repeats (MARc1, MARc2, MARc3, and MARc4) and three divergent repeats (MARa, MARb, and MARd), which play a vital role during the *Eimeria* invasion. *Enterococcus faecalis* is a native probiotic in animal intestines and can regulate intestinal flora. In this study, BC1 and C4D domains of EtMIC3, BC1 or C4D fusing to dendritic cells targeting peptides, were surface-displyed by *E. faecalis,* respectively. Oral immunizations were performed to investigate immune protective effects against *Eimeria* infection.

**KEYWORDS**  *Eimeria*, *Enterococcus faecalis*, microneme protein 3, immune protection

Address correspondence to Chunli Ma, machunli@neau.edu.cn, or Dexing Ma, madexing@neau.edu.cn.

The authors declare no conflict of interest.

See the funding table on p. 17.

Avian coccidiosis is a kind of protozoan disease caused by single or multiple *Eimeria* parasites that infect the intestinal epithelial cells of birds. *Eimeria tenella*, one of the most common and severe pathogenic *Eimeria* species, mainly parasitizes the cecum. Chickens infected by *E. tenella* display obvious clinical symptoms such as lethargy, anorexia, and bloody diarrhea, resulting in reduced feed utilization, body weight loss, and even death (1). The current controlling methods for avian coccidiosis mainly rely on in-feed anti-coccidial chemical drugs or the application of attenuated live vaccines, which consist of several parasites. However, outbreaks of coccidiosis on chicken farms all over the world still commonly occur. The direct or indirect side effects related to traditional controlling methods, including drug residues in chicken meat, drug resistance of *Eimeria* strains, and virulence reversion of live vaccines, encourage researchers to develop novel anti-coccidial vaccines (2). In the past few decades, much research work has focused on microneme proteins secreted by *E. tenella* with the aim of deeply understanding the pathogenic and immune mechanisms. Microneme proteins are secreted by microneme organelles, which include *E. tenella* microneme protein 2 (EtMIC2) (3), EtMIC3 (4, 5), EtMIC4 (6), EtMIC5 (7), and EtMIC8 (8). The EtMIC3 protein was coded by 988 amino acids that contained seven tandem microneme adhesion repeat regions (MARRs), named EtMIC3-MARa, EtMIC3-MARb, EtMIC3-MARc, and EtMIC3-MARd, respectively, among which MARa, MARb, and MARd are divergent external repeats; MARc consists of MARc1, MARc2, MARc3, and MARc4, which are four highly conserved internal repeats (4). Studies have shown that EtMIC3 specifically recognizes sialylated glycans and contributes to the tissue tropism of *E. tenella* to the ceca (9). A recent study demonstrated that the invasion of cecal epithelial cells by *E. tenella* sporozoites was partially blocked by anti-EtMIC3 serum (10). Oral immunization with recombinant attenuated *Salmonella typhimurium* delivering two domains of the EtMIC3 protein induced obvious immune responses and offered protective effects against homologous infection (11).

*Enterococcus faecalis*, a kind of native bacteria in animal intestines, can regulate intestinal flora and show probiotic functions. The reported study has shown that the genome sequence of *E. faecalis* contains functional genes related to intestinal adhesion and tolerance to acid, bile, and digestive enzymes (12). *E. faecalis* also displayed other characteristics, including antioxidant defenses and activating immune responses (13, 14). In addition, *E. faecalis* was developed as a vehicle to deliver heterologous antigens to stimulate systemic immune responses (15).

Fusing dendritic cells (DCs) targeting peptides (DCpep) with antigen is generally recognized as a promising strategy for improving the immunogen of the target protein. Previous studies have demonstrated that oral vaccination with recombinant *Lactobacillus* delivering DCpep and immunogenic antigens induced stronger antigen-specific cellular and humoral immune responses and provided more protection against pathogen infection (16–18). Our previous research also revealed that immunization of chicks with recombinant *E. faecalis* MDXEF-1-expressing fusion proteins of *E. tenella* 3-1E and DCpep evoked effective humoral and cellular immune responses and offered intensified anti-coccidial protections (15). The aim of the present study is to explore whether surface-anchored DCpep-BC1 or DCpep-C4D domains of EtMIC3 delivered by *E. faecalis* MDXEF-1 could stimulate effective cellular and humoral immune responses via oral administration, and whether the immune protective efficacies against homologous infection could be effectively enhanced.

## RESULTS

### Detection of C4D-specific antisera

In order to prepare C4D-specific antisera that are used to detect C4D protein expression in host bacteria *E. faecalis*, the C4D protein was first expressed in *E. coli* BL21. A target band of recombinant C4D-GST (rC4D-GST) protein was observed on sodium dodecyl sulfate-polyacrylamide gel electrophoresis (SDS-PAGE) gel with a molecular weight of about 53 kDa (Fig. 1A), which is equal to the molecule weight of GST tag protein in pGEX-6P-1 (26 kDa) and C4D protein (27 kDa). The rC4D-GST protein purified by the

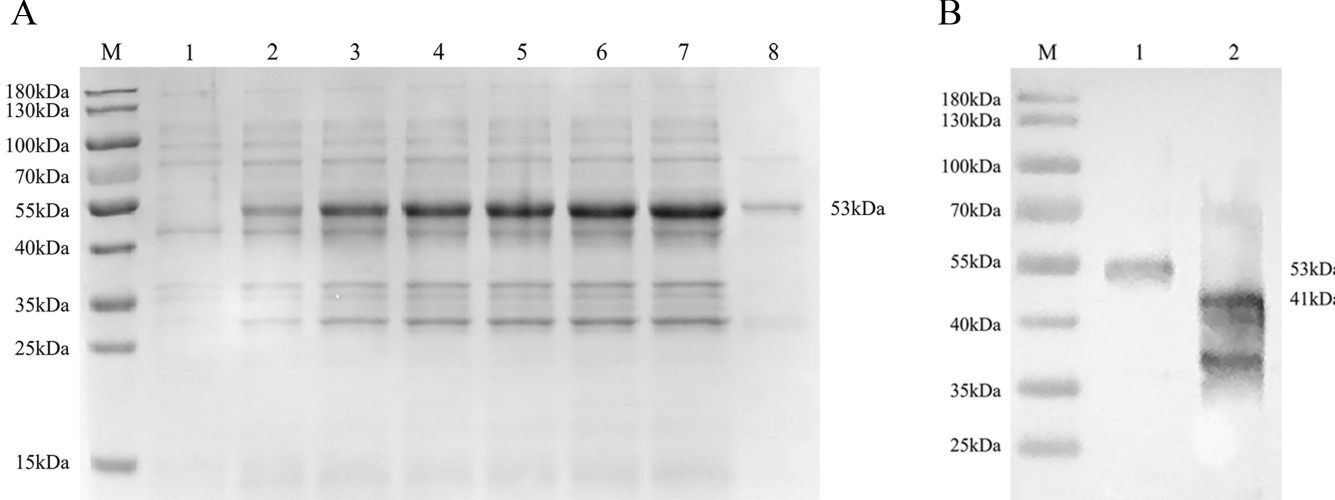

**FIG 1** Characterization of a specific antibody against the EtMIC3-C4D protein. C4D domain of *E. tenella* microneme protein 3 was expressed in *E. coli* BL21 cells, and an expected band of 53 kDa was observed (A). M, protein molecular weight marker. Lane 1, expressed bacteria protein in *E. coli*. BL21 cells without induction by IPTG (isopropyl β-D-thiogalactopyranoside; negative control). Lane 2–7, expressed EtMIC3-C4D protein in *E. coli*. BL21 cells that were induced by IPTG for 1, 2, 3, 4, 5, and 6 h, respectively. Lane 8, purified EtMIC3-C4D protein by affinity chromatography with Ni-conjugated sepharose. Detection of fusion proteins rC4D-GST (four repeat domain C and one domain D) and His6-BC1 (one domain B and one repeat domain C) by western blot using anti-C4D-specific polyclonal antisera as primary antibody (B). Lane 1, band of rC4D-GST fusion protein (53 kDa). Lane 2, His6-BC1 fusion protein (41 kDa).

GST-tagged protein purification kit (ThermoFisher Scientific, China) also showed a single band of about 53 kDa (Fig. 1A). The titer of the prepared C4D-specific polyclonal antisera reached $2^{16}$. Then, fusion proteins rC4D-GST (which contains four repeat domain C and one domain D) and His6-BC1 (which contains domain B and one repeat domain C) were detected with anti-C4D-specific polyclonal antisera using western blot, showing target bands of rC4D-GST fusion protein (53 kDa) (Fig. 1B) and His6-BC1 fusion protein (41 kDa) (Fig. 1B).

## Expression of target protein in *Enterococcus faecalis*

Plasmids pTX8048-SP-BC1-CWA, pTX8048-SP-DCpep-BC1-CWA, pTX8048-SP-C4D-CWA, and pTX8048-SP-DCpep-C4D-CWA were electrotransformed into MDXEF-1 host bacteria, and four recombinant *E. faecalis*, abbreviated as MDXEF-1/BC1-CWA, MDXEF-1/DC-BC1-CWA, MDXEF-1/C4D-CWA, and MDXEF-1/DC-C4D-CWA, were obtained. Four target fusion proteins expressed on the surface of positive bacteria were detected by western blot, showing about 71 kDa of cell wall-anchored DCpep-BC1 fusion protein (Fig. 2A) and BC1 (Fig. 2B) and 66 kDa of surface-anchored DCpep-C4D fusion protein (Fig. 2C) and C4D (Fig. 2D). Indirect immunofluorescence assay showed that fluorescence was observed on the surface of four recombinant bacteria: MDXEF-1/BC1-CWA (Fig. 2F), MDXEF-1/DC-BC1-CWA (Fig. 2G), MDXEF-1/C4D-CWA (Fig. 2I), and MDXEF-1/DC-C4D-CWA (Fig. 2J).

## Levels of sera IgG and sIgA in cecal lavage fluid

At 14 days after each immunization, levels of antigen-specific IgG in serum and sIgA in cecal lavage fluids from chickens in each group are shown in Fig. 3. The levels of IgG and sIgA in the groups MDXEF-1/BC1-CWA, MDXEF-1/DC-BC1-CWA, MDXEF-1/C4D-CWA, and MDXEF-1/DC-C4D-CWA gradually elevated along with the increase in immunization times, which were all higher than the phosphate-buffered saline (PBS) and MDXEF-1 groups ($P < 0.01$). The group immunized with MDXEF-1/DC-C4D-CWA and MDXEF-1/DC-BC1-CWA showed higher levels of IgG and sIgA than the MDXEF-1/C4D-CWA and MDXEF-1/BC1-CWA groups, respectively ($P < 0.01$). mRNA levels of cytokines in spleen tissues.

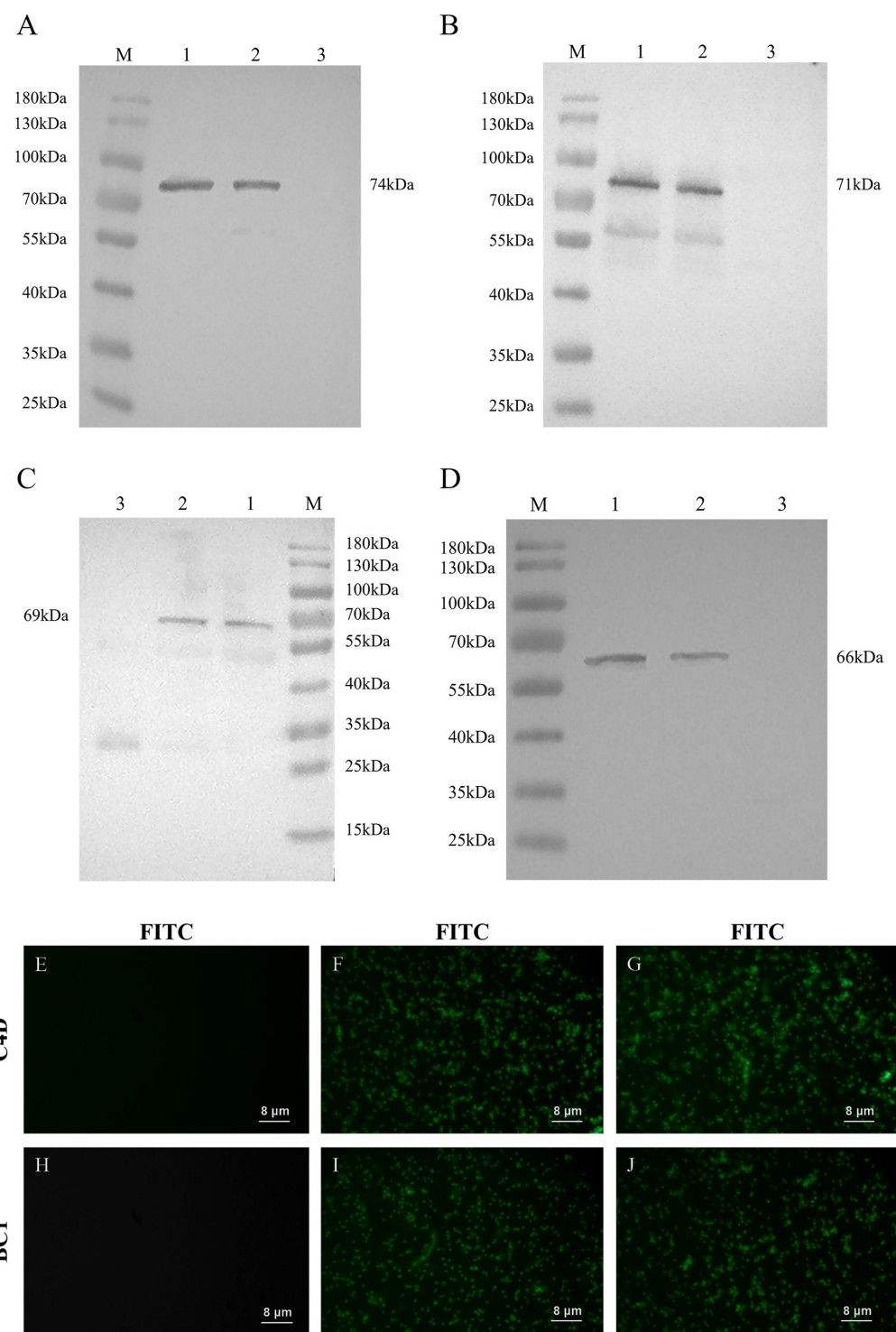

**FIG 2** Detection of cell wall-anchored protein in recombinant *E. faecalis*. Detection of cell wall-anchored protein by western blot for DCpep-BC1 (A), BC1 (B), DCpep-C4D (C), and C4D (D) and by indirect immunofluorescence for recombinant bacteria MDXEF-1/BC1-CWA (F), MDXEF-1/DC-BC1-CWA (G), MDXEF-1/C4D-CWA (I), and MDXEF-1/DC-C4D-CWA (J) using rabbit anti-BC1 or C4D polyclonal antisera as primary antibody. Lane M, protein molecular weight marker (A, B, C, D). Lanes 1 and 2, 71 kDa of cell wall-anchored DCpep-BC1 fusion protein (A), BC1 protein (B), 66 kDa of surface-anchored DCpep-C4D fusion protein (C), and C4D protein (D). Lane 3, negative control (cell wall-anchored protein of nisin-induced MDXEF-1/pTX8048). (E and H) Negative control (nisin-induced MDXEF-1/pTX8048).

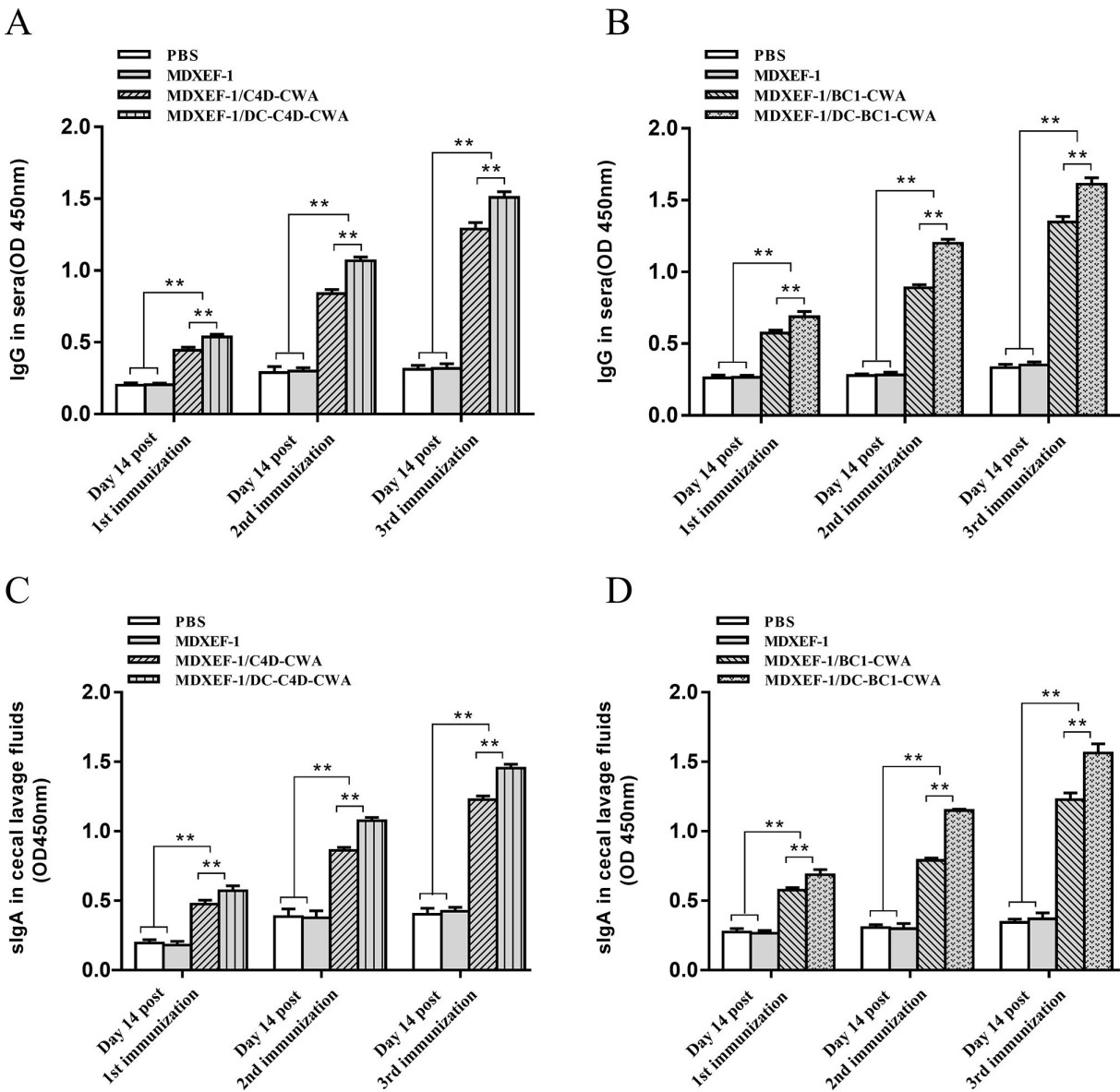

**FIG 3** Levels of antigen-specific IgG in serum and sIgA in cecal lavage fluids. At 14 days after each immunization, serum and cecal lavage fluids from chickens in each group were prepared. C4D or BC1 protein was used to coat a 96-well plate. Sera (1:300) or cecal lavage fluids (1:200) were used as primary antibodies, and horseradish peroxidase-conjugated goat anti-chicken IgG (1:2,000) or IgA (1:5,000) was applied as a secondary antibody, respectively. o-Phenylenediamine (1 mg/mL) and $H_2O_2$ (0.01%) were added, and the reaction was stopped by 2 M $H_2SO_4$. The absorbance was measured at 490 nm. Levels of IgG in sera (A and B) and sIgA in jejunal lavage fluid (C and D) are expressed as mean ± SD ($n = 3$). **$P < 0.01$.

As shown in Fig. 4, mRNA levels of ImL2, IFN-γ, ImL4, ImL10, IL-6, and IL-17 in spleen tissues from groups MDXEF-1/BC1-CWA, MDXEF-1/DC-BC1-CWA, MDXEF-1/C4D-CWA, and MDXEF-1/DC-C4D-CWA were all significantly upregulated with increasing immunization times and displayed higher antibody levels than those from the PBS and MDXEF-1 groups ($P < 0.01$). Moreover, the MDXEF-1/DC-C4D-CWA and MDXEF-1/DC-BC1-CWA groups were significantly higher than the MDXEF-1/C4D-CWA and MDXEF-1/BC1-CWA groups, respectively ($P < 0.01$).

## Proliferation of peripheral blood lymphocytes

The proliferation of peripheral blood lymphocytes (PBLs) was illustrated in Fig. 5. At 14 days after three immunizations, PBLs prepared from chickens immunized with MDXEF-1/

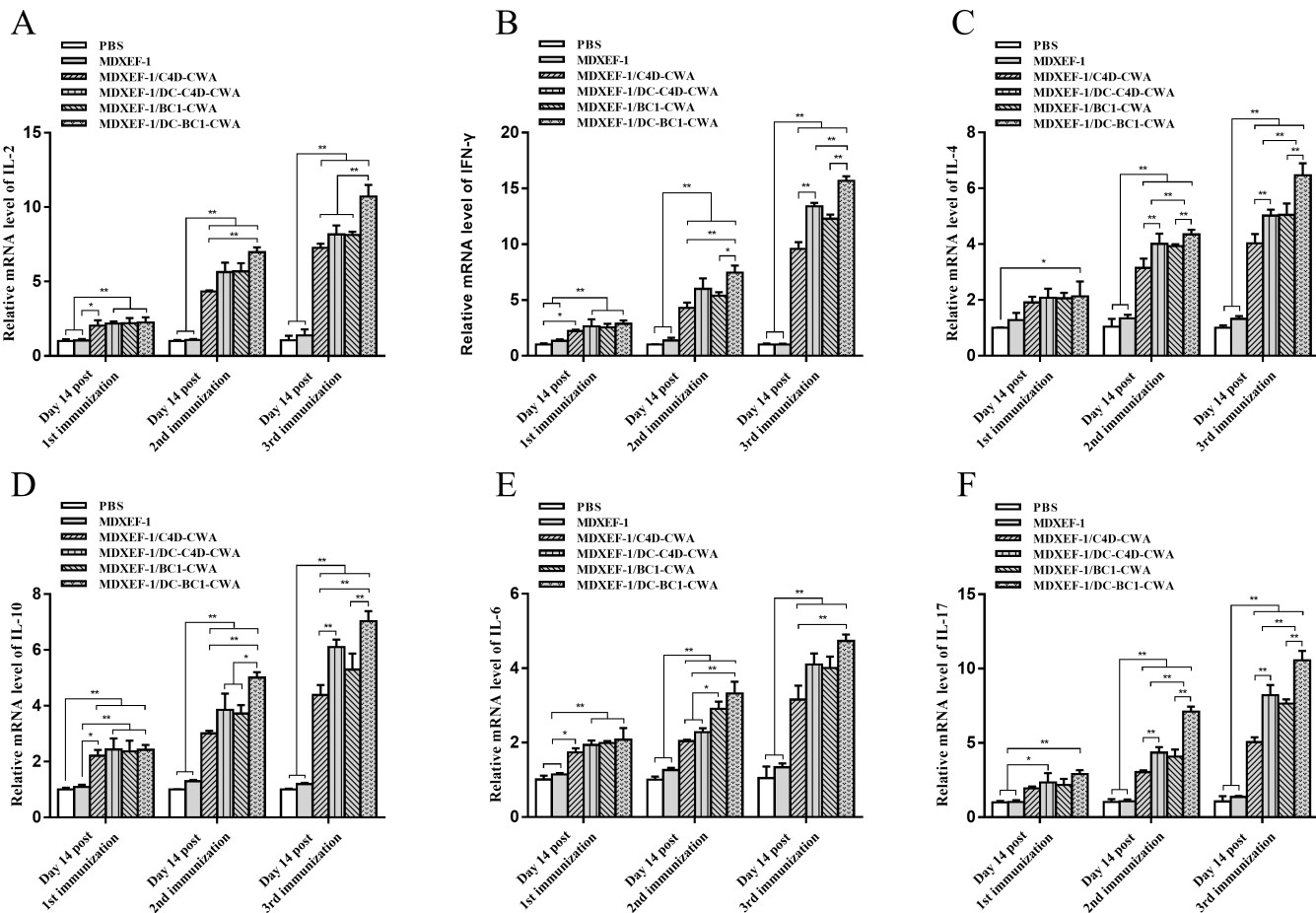

**FIG 4** mRNA expression levels of cytokines in spleen tissues. At 14 days after each immunization, the relative mRNA levels of chicken interleukin 2 (A), chicken interferon-γ (D), chicken IL-4 (B), chicken IL-10 (E), chicken IL-6 (C), and chicken IL-17 (F) in the spleen tissues of chickens ($n = 3$) from each group were determined by real-time quantitative reverse transcription PCR. β-Actin was used as a reference gene for the same sample. Data are expressed as mean ± SD ($n = 3$). *$P < 0.05$, **$P < 0.01$.

BC1-CWA and MDXEF-1/DC-BC1-CWA showed significantly higher proliferative responses to stimulation of BC1 proteins than those with the MDXEF-1 and PBS groups ($P < 0.01$), and the MDXEF-1/DC-BC1-CWA group was higher than the MDXEF-1/BC1-CWA group ($P < 0.01$). Similar proliferative responses to C4D protein were also observed in the groups immunized with MDXEF-1/C4D-CWA and MDXEF-1/DC-C4D-CWA, showing higher levels of antigen-specific proliferative responses than the PBS and MDXEF-1 groups ($P < 0.01$), and also the MDXEF-1/DC -C4D-CWA group was higher than the MDXEF-1/C4D-CWA group ($P < 0.01$). However, no significant differences in the proliferative responses to ConA stimulation were observed between groups immunized with protein-expressing bacteria and two control groups ($P > 0.05$).

## Body weight gain and oocyst reduction ratio

Body weight gain of chickens infected with *E. tenella* showed varying degrees of reduction (Table 1). No significant differences were observed in the weight gain of chickens among the four groups immunized with recombinant *E. faecalis* expressing target protein and the uninfected control group. In contrast, weight gains were significantly reduced in chickens in the infected control and vector control groups compared to the uninfected control group. As shown in Fig. 6, the oocyst reduction ratios in the groups MDXEF-1/BC1-CWA, MDXEF-1/DC-BC1-CWA, MDXEF-1/C4D-CWA, and MDXEF-1/DC-C4D-CWA were 43.50％, 50.85％, 49.15％, and 60.45％, respectively.

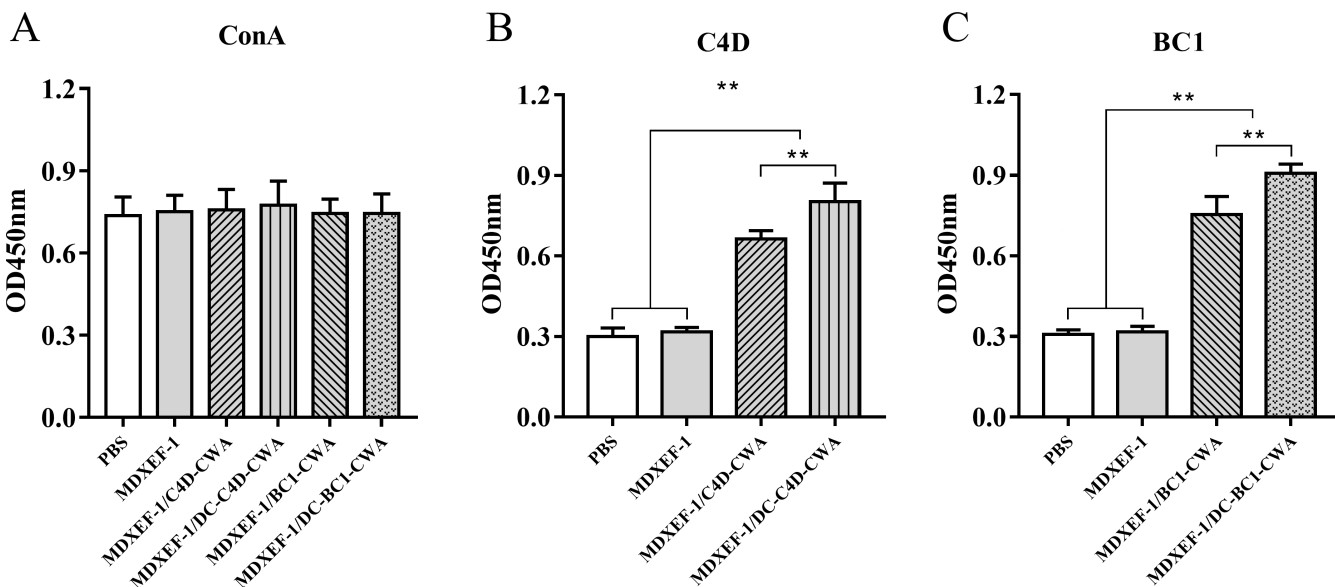

**FIG 5** Proliferation of PBLs. At 14 days after the third immunization, PBLs of chickens ($n$ = 3) from each group were prepared from peripheral blood. The proliferation of PBLs was assessed using the CCK-8 assay kit. ConA (A), C4D (B), and BC1 (C) proteins were selected as stimuli, respectively. Data are presented as mean ± SD ($n$ = 5). **$P$ < 0.01.

## Levels of inflammatory factors

On day 7 post-infection, protein levels of inflammatory factors interleukin 1 beta (IL-1β), IL-6, IL-8, and tumor necrosis factor-α (TNF-α) in the cecal tissues of chickens in each group were shown in Fig. 7. Levels of the four inflammatory factors in the groups MDXEF-1/BC1-CWA, MDXEF-1/DC-BC1-CWA, MDXEF-1/C4D-CWA, and MDXEF-1/DC-C4D-CWA were remarkably lower than those in the infected control and vector control groups ($P$ < 0.01). Notably, groups MDXEF-1/DC-C4D-CWA and MDXEF-1/

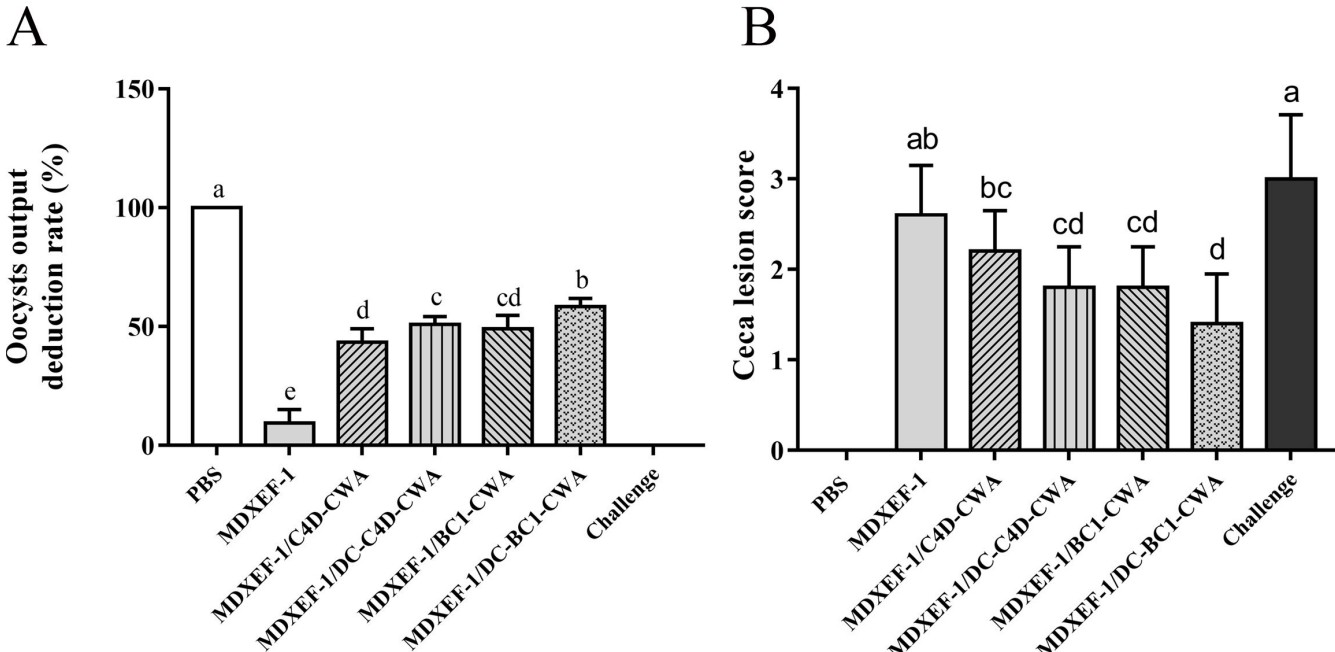

**FIG 6** Cecal lesion score and oocyst decrease ratio. From 7 to 11 days post-infection, chickens ($n$ = 10) per group were used for oocyst counting (A). At 1 week post-challenge, chickens ($n$ = 5) per group were randomly selected for evaluating cecal lesion scores (B). Different small letters mean significance ($P$ < 0.05).

**TABLE 1** Changes in the body weight of chicks in each group[a]

| Groups | Average weight gain (g) | Relative weight gain (%) |
|---|---|---|
| Challenge | $61.19 \pm 28.11^c$ | 55.04 |
| MDXEF-1/pTX8048 | $70.37 \pm 16.83^{bc}$ | 63.30 |
| MDXEF-1/C4D-CWA | $94.15 \pm 29.19^{abc}$ | 84.69 |
| MDXEF-1/DC-C4D-CWA | $100.81 \pm 30.15^{ab}$ | 90.68 |
| MDXEF-1/BC1-CWA | $99.46 \pm 27.63^{ab}$ | 89.47 |
| MDXEF-1/DC-BC1-CWA | $103.84 \pm 44.22^{ab}$ | 93.41 |
| PBS | $111.17 \pm 38.07^a$ | 100 |

[a]A significant difference ($P < 0.05$) was shown by different small letters (a, b, c) between two numbers in a column.

DC-BC1-CWA were significantly lower than MDXEF-1/C4D-CWA and MDXEF-1/BC1-CWA, respectively ($P < 0.01$).

## Pathological changes

On day 7 post-infection, the ceca of chickens from the challenged control group showed typical gross pathological lesions, including swelling and thickening of the cecal wall, hemorrhage on the serous membranes, and bloody cecal content. The lesion scores in the ceca of chickens in the groups MDXEF-1/BC1-CWA, MDXEF-1/DC-BC1-CWA, MDXEF-1/C4D-CWA, and MDXEF-1/DC-C4D-CWA were not as severe as those in the infected control group ($P < 0.05$) (Fig. 8). Histopathological changes in the cecal tissues of chickens in the infected control and vector control groups displayed obvious inflammatory cell infiltration, loss of cecal villi, a broken structure of the gland in the lamina propria, and a large number of blood cells and oocysts in the submucosa. By contrast, the histopathological injury in the cecal tissues of chickens immunized with MDXEF-1/BC1-CWA, MDXEF-1/DC-BC1-CWA, MDXEF-1/C4D-CWA, and MDXEF-1/DC-C4D-CWA was all relatively slight (Fig. 9).

## DISCUSSION

Avian coccidiosis caused by *Eimeria* brings huge economic losses to the poultry industry. Live vaccines and anti-coccidial drugs have been widely used for a long time. The development of novel, safe, effective, and economically applied vaccines or biological preparations has already become a research hotspot, mainly because of the incidence of drug-resistant strains of *Eimeria* species, drug residues in chicken products, and the virulence reversion of live vaccines. *Eimeria* parasites have complex life cycles, including oocyst, sporozoite, trophozoite, schizont, and gametophyte, and parasites in each stage contain a variety of antigenic components. Therefore, an effective and representative antigen is particularly critical to developing anti-coccidial subunit vaccines. Microneme proteins and surface antigen proteins are considered immunodominant antigens for developing anti-coccidial vaccines (19–22). Microneme proteins (MICs), secreted from microneme organelles that are located at the parasite apex, are widely reported to interact with specific receptors on the surface of intestinal epithelial cells and play an important role during *Eimeria* adhesion and invasion (9, 10). In addition, previous studies have shown chickens orally administered with attenuated *Salmonella typhimurium* expressing two MAR domains at the C-terminal of the EtMIC3 protein displayed partial anti-coccidial effects (11). Our previous study showed that peptides specifically bound to BC1 adhesion structures of the EtMIC3 protein could inhibit the invasion of sporozoites into MDBK cells, showing an inhibition rate of up to 71.8% (23). In this study, BC1 and C4D domains, two representative repeat structures of the EtMIC3 protein, were selected as target antigens to investigate immune protective effects against homologous infection. Up to now, many kinds of vehicles, including *Bacillus subtilis* (24), *Enterococcus faecalis* (25), PLGA nanospheres (26), *Saccharomyces cerevisiae* yeast (27), and *Lactobacillus plantarum* (28), were applied to deliver candidate antigens or peptides to explore the immune prevention of *Eimeria* infection. Our previous research has focused on the exploration of using *E. faecalis* as carrier vectors and showed that fusion of DCpep

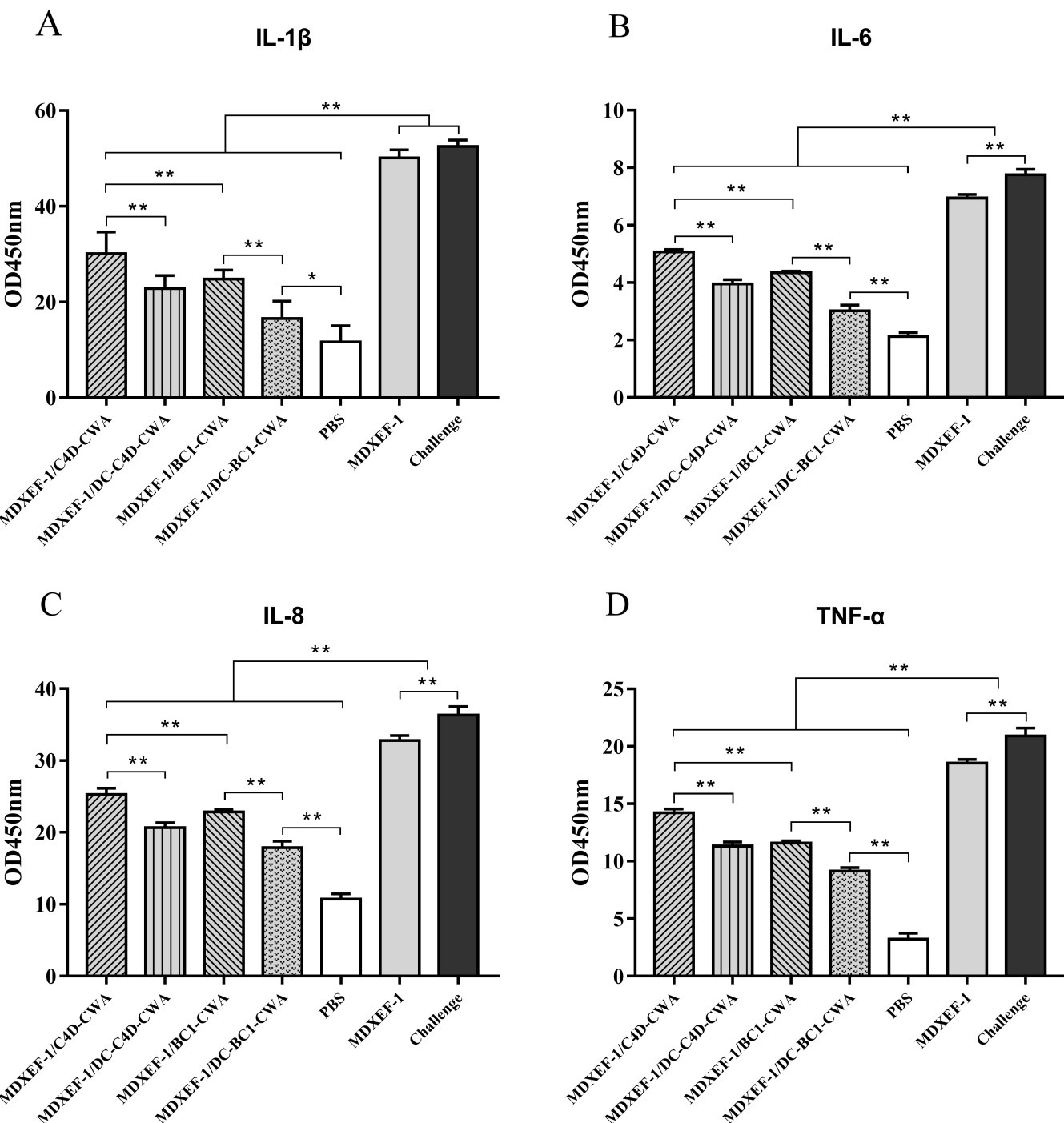

**FIG 7** Levels of inflammatory cytokines in the cecal tissues. On day 7 post-infection, the protein levels of inflammatory cytokines, including IL-1β (A), IL-6 (B), IL-8 (C), and TNF-α (D), in the cecal tissues of chickens ($n = 3$) in each group were determined using the enzyme-linked immunosorbent assay method. Data are presented as mean ± SD ($n = 5$). *$P < 0.05$, **$P < 0.01$.

with candidate antigen delivered by *E. faecalis* enhanced the antigen-specific immune response (15), indicating that the introduction of DCpep could improve the presentation of surface-displayed fusion antigen to dendritic cells and then enhance the immune response against homologous infection. In the present study, DCpep was fused to the C terminal of target antigen BC1, C4D to generate DCpep-BC1 or DCpep-C4D, which was

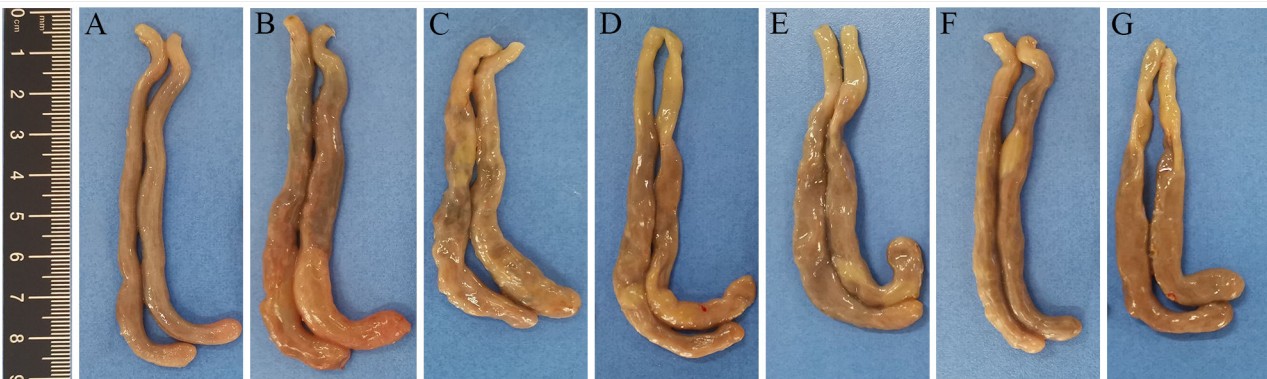

**FIG 8** Gross pathological changes in the ceca of chickens. The ceca of chickens from the challenged control (B) and vector control groups (C) showed typical gross pathological lesions, including swelling and thickening of the cecal wall, hemorrhage on the serous membranes, and bloody cecal content. The lesion scores in the ceca of chickens in the groups MDXEF-1/C4D-CWA (D), MDXEF-1/DC-C4D-CWA (E), MDXEF-1/BC1-CWA (F), and MDXEF-1/DC-BC1-CWA (G) were not as severe as those in the infected control group. There are no apparent lesions in the ceca of chickens in the negative control (A).

then subcloned into downstream cell wall-anchored (CWA) sequences in the expressing vector pTX8048 to express the fusion protein on the surface of recombinant *E. faecalis*.

Previous studies have shown that humoral immunity plays a vital role in resistance to coccidial infection (29). In this study, the levels of antigen-specific IgG in sera and sIgA in cecal lavage fluids elicited by the four recombinant *E. faecalis* expressing BC1 or C4D proteins were higher than those in the PBS and vector groups ($P < 0.01$), indicating that recombinant bacteria indeed stimulate significant humoral immune responses. Moreover, BC1 or C4D protein fused with DCpep induced higher levels of IgG and sIgA than BC1 or C4D protein alone delivered by *E. faecalis*, respectively ($P < 0.05$), indicating that DCpep effectively improved the target delivery of BC1 or C4D antigens to dendritic cells, increased uptake and antigen presentation, and offered more protective efficacies against homologus challenge, which are consistent with previous reports (11, 21).

Cellular immunity was generally accepted to play a critical role in anti-coccidial infection, and both CD4+αβ and CD8+γδ T cells were recruited to the site of *Eimeria*

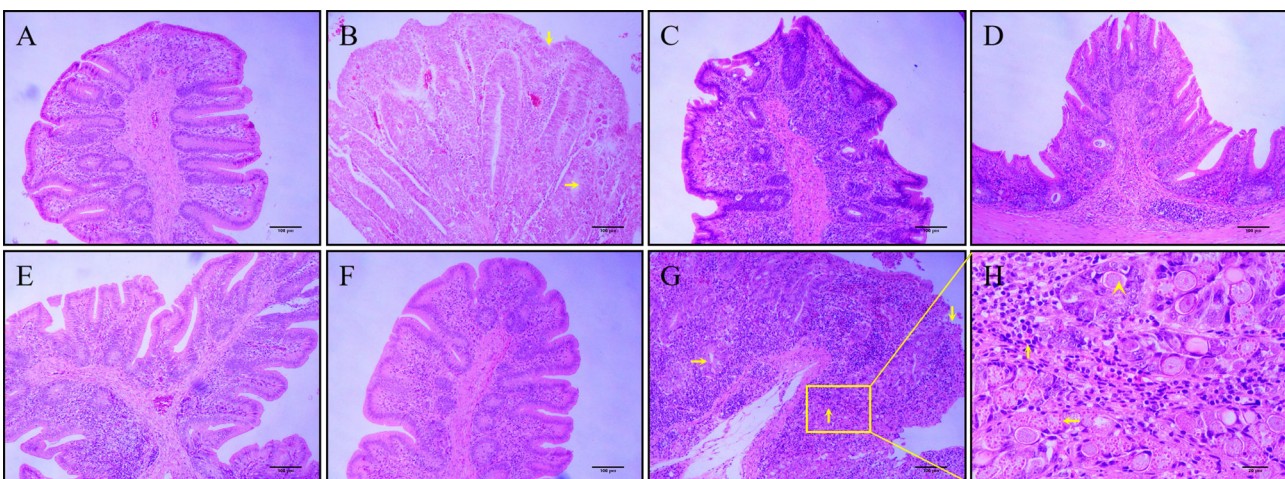

**FIG 9** Histopathological changes in the ceca of chickens. At 7 days post-infection, the cecal tissues of chickens in each group were sampled and fixed in 10% formalin, embedded with paraffin, sectioned, and stained with hematoxylin and eosin. Histopathological changes in the cecal tissues of chickens in the infected control (G and H) and vector control groups (B) displayed obvious infiltration of inflammatory cells (↑), destroyed cecal villi (↓), broken structure of gland in lamina propria (→), large number of blood cells and oocysts (◄), and macrogametocytes (←) in submucosa. By contrast, the histopathological injury in the cecal tissues of chickens immunized with MDXEF-1/C4D-CWA (C), MDXEF-1/DC-C4D-CWA (D), MDXEF-1/BC1-CWA (E), and MDXEF-1/DC-BC1-CWA (F) was all relatively slight. No histopathological changes were observed in the ceca of chickens from the negative control (A).

infection (30, 31). In this study, the groups immunized with four recombinant *E. faecalis* expressing BC1 or C4D, especially BC1-DCpep or C4D-DCpep fusion protein, showed higher proliferation of PBLs and higher levels of cytokines IL-2, IFN-γ, IL-4, IL-10, IL-6, and IL-17 than the PBS and vector control groups (*P* < 0.01), which indicated that BC1 or C4D proteins delivered by *E. faecalis* effectively activated cell-mediated immunity. A previous report demonstrated that IL-2 and IFN-γ play a vital role in immune protection against *Eimeria* infection (32), and IL-4 and IL-10 contribute to stimulating the proliferation of B lymphocytes (33). Furthermore, Th17 cells participate in innate immune responses and inflammatory responses by secreting cytokines such as IL-17 and TNF-α and therefore play an important role in anti-coccidial infection (34, 35). The above results suggested that the pathogens could be effectively eliminated by activating cellular immunity. Notably, cell wall-anchored BC1 and C4D proteins fused by DCpep induced stronger cellular immunity than BC1 and C4D proteins, respectively, indicating that DCpep further enhanced specific cellular immune responses. The above results are consistent with the results of animal testing, showing that chickens immunized with recombinant *E. faecalis* displaying surface-anchored BC1, C4D, C4D-Dcpep, and BC1-Dcpep, especially the BC1-DCpep fusion protein, showed significantly higher weight gain, lower oocyst excretion, lower cecal lesion scores, and relatively mild gross pathological lesions and histopathological changes in the ceca compared with the infected control group.

Previous studies have demonstrated that the expression levels of inflammatory factors such as IL-1β, IL-6, IL-8, and TNF-α are significantly increased after coccidial infection (36). The results of this study showed that the levels of IL-6, IL-8, IL-1β, and TNF-α were significantly decreased in the four groups immunized with recombinant *E. faecalis* compared with the infected and vector control groups. It is worth noting that cell wall-anchored fusion proteins BC1-DCpep and C4D-DCpep induced lower levels of inflammatory factors than cell wall-anchored BC1 or C4D proteins delivered by *E. faecalis*, respectively (*P* < 0.05). The above results indicated that recombinant *E. faecalis* expressing surface-anchored BC1 or C4D protein could effectively relieve intestinal inflammatory injury, which could be enhanced by fusing with DCpep. These results may be explained by the fact that the introduction of DCpep increases targeting and subsequent uptake by dendritic cells, stimulates stronger humoral and cellular immune responses, and reduces inflammatory damage to the cecal tissues caused by *Eimeria* parasites. In addition, the protective efficacy offered by the delivered BC1 protein was superior to that of the C4D protein, which coincides with the dynamic changes of inflammatory factors and may be explained by the differences in antigenicity of the relatively unconserved domains B and D among the seven repeat structural domains of the EtMIC3 protein.

## Conclusion

Overall, oral immunization with recombinant *E. faecalis* delivering cell wall-anchored BC1 or C4D proteins fused or non-fused DCpep induced antigen-specific cellular and humoral immune responses and provide partial immune protection against homologous infection. The fusion expression of the target protein and DCpep further enhanced the antigen-specific immune effects. This study provides a reference for the exploration of *E. faecalis* vaccine based on the EtMIC3 protein.

## MATERIALS AND METHODS

### Bacteria strains, parasites, and animals

The strains and plasmids used in this study are listed in Table 2. The bacteria strain *E. faecalis* MDXEF-1 was cultured at 30°C without shaking in GM17 culture medium, which consists of M17 medium (Luqiao, Beijing) and 0.5% glucose. The bacteria *E. coli* was cultured in Luria-Bertani medium at 37°C with shaking. The polyclonal antisera against EtMIC3 BC1 protein was prepared and preserved in our laboratory (23). The EtMIC3-C4D gene (KM018021.1) fragment was amplified and stored in our laboratory.

**TABLE 2** Strains and plasmids used in this study

| Strains or plasmids | Relevant characteristics | Source |
|---|---|---|
| Strains | | |
| *E. coli* DH5α | SupE441lacU169(φ80 lacZ1M15) hsdR17 recA1 endA1 gyrA96 thi-1 relA1, plasmid-free | TaKaRa, China |
| *E. coli* BL21 | F-ompT hsdSB (rB− mB−) gal dcm (DE3), plasmid-free | TaKaRa, China |
| *E. coli* BL21/pET30a-BC1 | With plasmid pET30a-BC1 in *E. coli* BL21 | (14) |
| *E. coli* BL21/pGEX-6P-1 | With plasmid pGEX-6P-1 in *E. coli* BL21 | This study |
| *E. coli* BL21/pGEX-C4D | With plasmid pGEX-C4D in *E. coli* BL21 | This study |
| *E. faecalis* MDXEF-1 | Isolated from chicken and reserved by our laboratory (Chinese patent ZL201410817717.5) | Stored in our laboratory |
| MDXEF-1/pTX8048 | With plasmid pTX8048 in *E. faecalis* MDXEF-1 | This study |
| MDXEF-1/BC1-CWA | With plasmid pTX8048-SP-BC1-CWA in *E. faecalis* MDXEF-1 | This study |
| MDXEF-1/DC-BC1-CWA | With plasmid pTX8048-SP-DCpep-BC1-CWA in *E. faecalis* MDXEF-1 | This study |
| MDXEF-1/C4D-CWA | With plasmid pTX8048-SP-C4D-CWA in *E. faecalis* MDXEF-1 | This study |
| MDXEF-1/DC-C4D-CWA | With plasmid pTX8048-SP-DCpep-C4D-CWA in *E. faecalis* MDXEF-1 | This study |
| Plasmid | | |
| PUC57-BC1 | With fragment encoding BC1 | (14) |
| PUC57-C4D | With fragment encoding C4D | Sangon Biotech |
| pGEX-6p-1 | *Escherichia coli* expression vector | This study |
| pGEX-C4D | With fragment encoding C4D protein in pET30a | This study |
| pTX8048 | With fragment encoding signal peptide of secretion protein Usp45 (SP) | (15) |
| pTX8048-SP-ΔHexon CWA | With fragment encoding signal peptide of secretion protein Usp45 (SP) and ΔHexon protein in anchored form, no dendritic cell-targeting peptides | (10) |
| pTX8048-SP-DC-ΔHexon-CWA | With fragment encoding signal peptide of secretion protein Usp45 (SP) and ΔHexon protein in anchored form, contains dendritic cell-targeting peptides | (10) |
| pTX8048-SP-BC1-CWA | With fragment encoding signal peptide of secretion protein Usp45 (SP) and BC1 protein in anchored form, no dendritic cell-targeting peptides | This study |
| pTX8048-SP-DC-BC1-CWA | With fragment encoding signal peptide of secretion protein Usp45 (SP) and BC1 protein in anchored form, contains dendritic cell-targeting peptides | This study |
| pTX8048-SP-C4D-CWA | With fragment encoding signal peptide of secretion protein Usp45 (SP) and C4D protein in anchored form, no dendritic cell-targeting peptides | This study |
| pTX8048-SP-DC-C4D-CWA | With fragment encoding signal peptide of secretion protein Usp45 (SP) and C4D protein in anchored form, contains dendritic cell-targeting peptides | This study |

Sporulated oocysts of *E. tenella* were stored in a 2.5% potassium dichromate solution at 4°C in our laboratory and passed by challenging chickens at least every 6 months. Male New Zealand rabbits weighing about 2.5 kg were obtained from a farm near Harbin, Heilongjiang Province. Newly hatched, specific-pathogen-free (SPF) White Leghorn chickens were purchased from Harbin Veterinary Research Institute (Harbin, Heilongjiang, China). Each chicken was fed a coccidiostat-free diet with water *ad libitum*.

## Expression of the EtMIC3-C4D protein

The EtMIC3-C4D gene fragment amplified by primer pairs C4D-F1 and C4D-R1 (Table 3) was cleaved by endonuclease *Bam*HI and *Kpn*I and purified using the TIANgel Midi Purification Kit (TIANGEN, Beijing, China). The purified C4D gene fragment was cloned into the *Bam*H I/*Kpn* I site of plasmid pGEX-6P-1 to generate pGEX-C4D, which was then transformed into *E. coli* BL21-competent cells. The screened-positive bacteria were induced by isopropyl β-D-thiogalactopyranoside (Solarbio, Beijing, China) with a final concentration of 1.0 mM. The expression of recombinant C4D protein fused with the GST tag (rC4D-GST) was identified by SDS-PAGE. The target protein rC4D-GST in the supernatant of ultrasonic cell lysate was purified using the GST-tag Purification Kit (Beyotime Biotechnology, Shanghai, China).

**TABLE 3** Primer sequences with their corresponding PCR product size[a]

| Genes | Name of primers | Primer sequences (5′–3′) | Enzyme | Product length |
|---|---|---|---|---|
| C4D | C4D/F1 | CGCGGATCCCAAGCTGTTCCTGAAGCA | *Bam*HI | 738 bp |
| (KM018021.1) | C4D/R1 | CCGCTCGAGTTATAAAGTAGCACGTTCACCCATTTTA | *Xho*I | |
| | C4D/*R2* | CGGGGTACCTAAAGTAGCACGTTCACCCATTTTA | *Kpn*I | |
| BC1 | BC1/F1 | CGCGGATCCACCCTGCAGGAAGCGCTGG | *Bam*HI | 864 bp |
| (KM018021.1) | BC1/R1 | CGGGGTACCGCTACCGCTCGGATCTTCG | *Kpn*I | |
| β-Actin | β-Actin/F | GCCAACAGAGAGAAGATGACAC | /[b] | 140 bp |
| (NM_205518.2) | β-Actin/R | GTAACACCATCACCAGAGTCCA | / | |
| IL-2 | IL-2/F | GTGGCTAACTAATCTGCTGTCC | / | 105 bp |
| (NM_204153.2) | IL-2/R | GTAGGGCTTACAGAAAGGATCAA | / | |
| IFN-γ | IFN-γ/F | CAAAGCCGCACATCAAACA | / | 80 bp |
| (NM_205149.2) | IFN-γ/R | TTTCACCTTCTTCACGCCATC | / | |
| IL-4 (NM_001007079.2) | IL-4/F | CTGTGCCCACGCTGTGCTTA | / | 83 bp |
| | IL-4/R | GGAAACCTCTCCCTGGATGTCA | / | |
| IL-10 (NM_001004414.4) | IL-10/F | CATGCTGCTGGGCCTGAA | / | 94 bp |
| | IL-10/R | CGTCTCCTTGATCTGCTTGATG | / | |
| IL-6 | IL-6/F | ATGGTGATAAATCCCGATGAAG | / | 153 bp |
| (NM_204628.2) | IL-6/R | CCTCACGGTCTTCTCCATAAAC | / | |
| IL-17 | IL-17/F | CCATTCCAGGTGCGTGAACT | / | 130 bp |
| (HQ008777.1) | IL-17/R | TTTCTTCTCCAGGCGGTACG | / | |

[a]The underlined sequence indicates restriction enzyme recognition.
[b]"/" means no enzyme site was contained in primer sequences.

## Characterization of polyclonal antisera against EtMIC3-C4D

Rabbit antisera against the rC4D-GST protein was prepared according to the reported method (30). Briefly, 2 mg of rC4D-GST fusion protein mixed with 2 mL of complete Freund's adjuvant was injected subcutaneously into the back of a New Zealand white rabbit. In the subsequent booster injection, anincomplete Freund's adjuvant was used. One week after the last injection, peripheral blood was collected via ear veins to prepare anti-rC4D sera. The titer of anti-rC4D sera was determined by enzyme-linked immunosorbent assay (ELISA) (15). In brief, a 96-well plate coated with 100 µL of rC4D-GST protein (10 µg/mL) in each well was incubated overnight at 4°C. After washing with PBS containing 0.05% Tween-20 (PBST), the plate was blocked with 5% skim milk in PBST solution (pH 7.2). Then, 00 µL of twofold serially diluted anti-rC4D sera was added to each well and incubated for 1 h at 37°C. After washing, the plate was incubated with horseradish peroxidase (HRP)-conjugated goat anti-rabbit IgG (1:5,000, Sigma-Aldrich) for 1 h at 37°C. Then, 100 µL of o-phenylenediamine (1 mg/mL) and $H_2O_2$ (0.01%) were added per well, and the reaction was stopped by 2 M $H_2SO_4$. The OD values were measured at 490 nm using a reader (Bio-Rad, USA).

The purified BC1-His6 fusion protein expressed in recombinant bacteria *E. coli* BL21/pET30a-BC1 (23), which also contains the same conserved domain C as the C4D protein, was used to detect the specificity of anti-rC4D sera by using western blot as previously described (23). Briefly, fusion proteins rC4D-GST and rBC1-His6 separated by SDS-PAGE were transferred to nitrocellulose membranes (Bio-Rad, Hercules, CA, USA). The membrane was blocked in Tris-buffer saline Tween-20 (TBST) solution containing 5% skimmed milk for 1.5 h. After washing with TBST, the membranes loaded with rC4D-GST and rBC1-His6 proteins were reacted with anti-rC4D-GST (1:2,000) rabbit serum overnight at 4°C. After washing, the membranes were incubated with HRP-conjugated goat anti-rabbit IgG (1:3,000, Sigma Aldrich) for 1 h at 37°C. The reaction was revealed by using the enhanced chemiluminescence (ECL) detection kit (Beyotime Biotechnology, Shanghai, China).

## Construction of a recombinant *Enterococcus faecalis* expressing target protein

The BC1 and C4D fragments were amplified with primer pairs BC1-F1/R1 and C4D-F1/*R2* (Table 2)**,** and restriction enzyme sites *Bam*HI and *Kpn*I were introduced at the 5′ and 3′ ends of the BC1 and C4D primer sequences, respectively. The target gene fragments BC1 and C4D and plasmids pTX8048-SP-ΔHexon-CWA and pTX8048-SP-DCpep-ΔHexon-CWA (16) were double digested with *Bam*HI and *Kpn*I. Then, the digested fragments BC1 and C4D were cloned into *Bam*H I/*Kpn* I sites of pTX8048-SP-DCpep-CWA, which contain DCpep and CWA sequences, to generate plasmids pTX8048-SP-C4D-CWA, pTX8048-SP-DCpep-C4D-CWA, pTX8048-SP-BC1-CWA, and pTX8048-SP-DCpep-BC1-CWA. The schematic diagram for the construction of plasmids is shown in Fig. 10. The above four identified plasmids were electrotransformed into *E. faecalis* MDXEF-1 by using Gene Pulser apparatus (Bio-Rad, Hercules, CA, USA). The four recombinant-positive *E. faecalis* were named MDXEF-1/BC1-CWA, MDXEF-1/DC-BC1-CWA, MDXEF-1/C4D-CWA, and MDXEF-1/DC-C4D-CWA.

## Western blot analysis

To detect the expression of BC1 and C4D proteins, the recombinant-positive *E. faecalis* were induced with nisin (Sigma Aldrich) as previously described (16). Briefly, the four recombinant *E. faecalis* expressing target protein and the control *E. faecalis* MDXEF-1 harboring plasmid pTX8048 were cultured in GM17 medium, and nisin (Sigma-Aldrich)

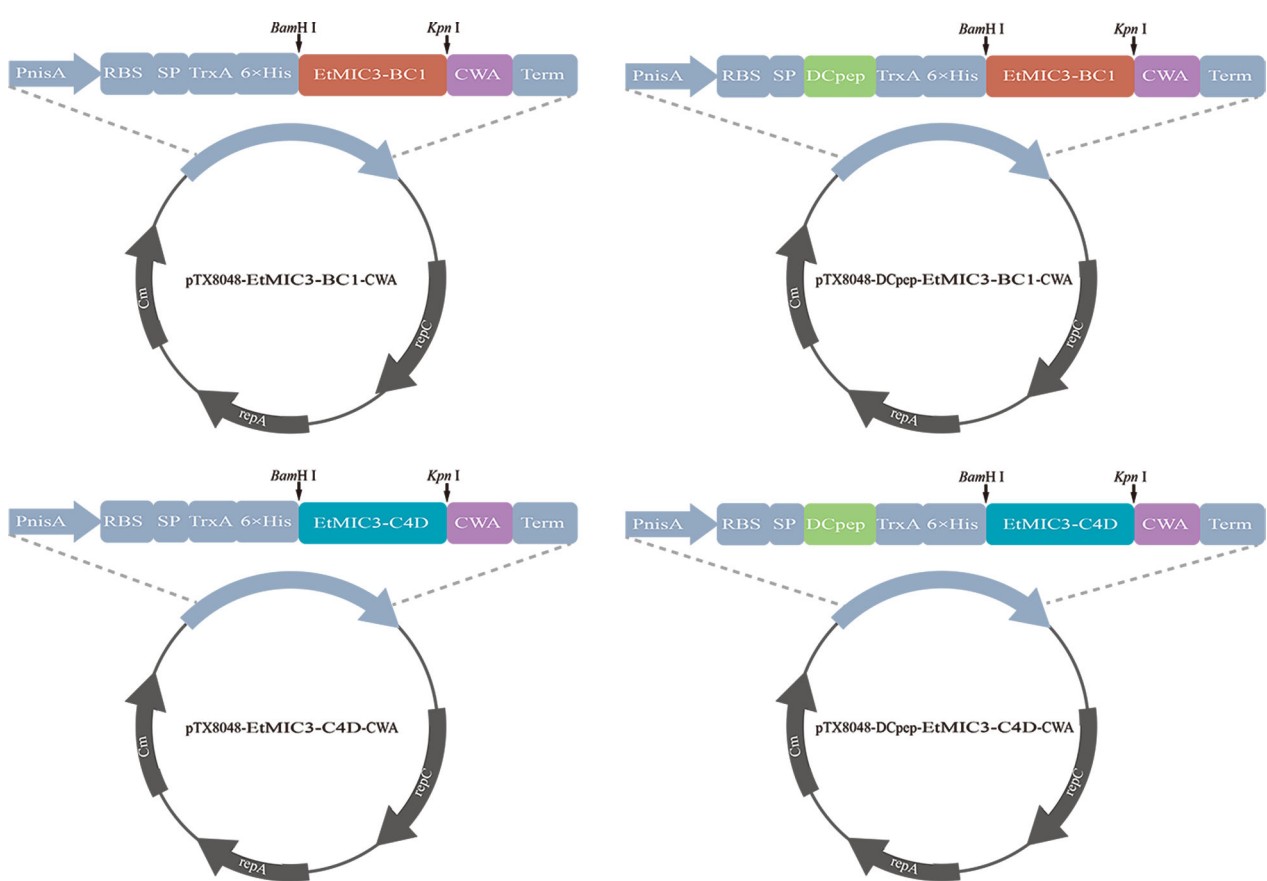

**FIG 10** Schematic diagram of plasmids pTX8048-SP-C4D-CWA, pTX8048-SP-BC1-CWA, pTX8048-SP-DCpep-C4D-CWA, and pTX8048-SP-DCpep-BC1-CWA. All the plasmids contain gene fragments of the nisin-inducible promoter, signal peptide from the Usp45 protein of *Lactococcus lactis* NZ9000, and CWA fused to the C-terminal of the C4D and BC1 domains of EtMIC3. Plasmids pTX8048-SP-DCpep-C4D-CWA and pTX8048-SP-DCpep-BC1-CWA contained a fused DCpep at the N-terminal of the C4D and BC1 proteins.

was added to the culture medium with a final concentration of 5 ng/mL. After inducing for 5 h, the cultured bacteria were harvested to prepare protein samples. The cell wall-anchored proteins BC1 and C4D displayed on the surface of bacteria were prepared as previously described (20). The prepared protein samples were separated by 10% SDS-PAGE, transferred to nitrocellulose membranes, and then blocked with 5% skim milk for 2 h at 37°C. The membranes were then incubated with rabbit anti-BC1 or anti-C4D polyclonal antisera (1:2,000) overnight at 4°C. The washed membranes were reacted with HRP-conjugated goat anti-rabbit IgG antibody (1:2,500, Sigma-Aldrich). The expected bands were visualized by the ECL detection system according to the provided procedures.

## Indirect immunofluorescent detection

To examine the surface-anchored target protein, recombinant bacteria MDXEF-1/BC1-CWA, MDXEF-1/DC-BC1-CWA, MDXEF-1/C4D-CWA, and MDXEF-1/DC-C4D-CWA were cultured in GM17 medium with 10 µg/mL chloramphenicol. When $OD_{600}$ values reached 0.5, nisin was added to the culture medium as described in previous studies (15) and cultured for 5 h at 37°C. After centifugation, the pellets were washed twice with sterile PBS (pH 7.2) and then incubated with rabbit polyclonal antisera against BC1 (1:200) or C4D (1:200) proteins. Fluorescein isothiocyanate-conjugated goat anti-rabbit IgG (1:50) (Solarbio, Beijing, China) was used as a secondary antibody. After reaction, the fluorescence on the surface of recombinant *E. faecalis* was observed using a fluorescence microscope (Leica DM2000).

## Immunization design

Animal grouping, immunizations, and challenges are displayed in Fig. 11. One-week-old SPF chicks were randomly divided into seven groups with 24 chickens each. Briefly, chickens in the PBS control and infection control groups were orally gavaged with 100 µL of PBS (pH 7.2), an empty vector control group with MDXEF-1/pTX8048, and the other four groups with recombinant *E. faecalis* MDXEF-1/BC1-CWA, MDXEF-1/DC-BC1-CWA, MDXEF-1/C4D-CWA, and MDXEF-1/DC-C4D-CWA. The immunization procedures included three phases: primary immunization (once a day and a total of three immunizations from days 7 to 9), secondary immunization (from days 21 to 23), and tertiary immunization (from days 35 to 37), each orally gavaged with $1.0 \times 10^{10}$ CFU per chicken. At 2 weeks after tertiary immunization, all chickens except the PBS control group were

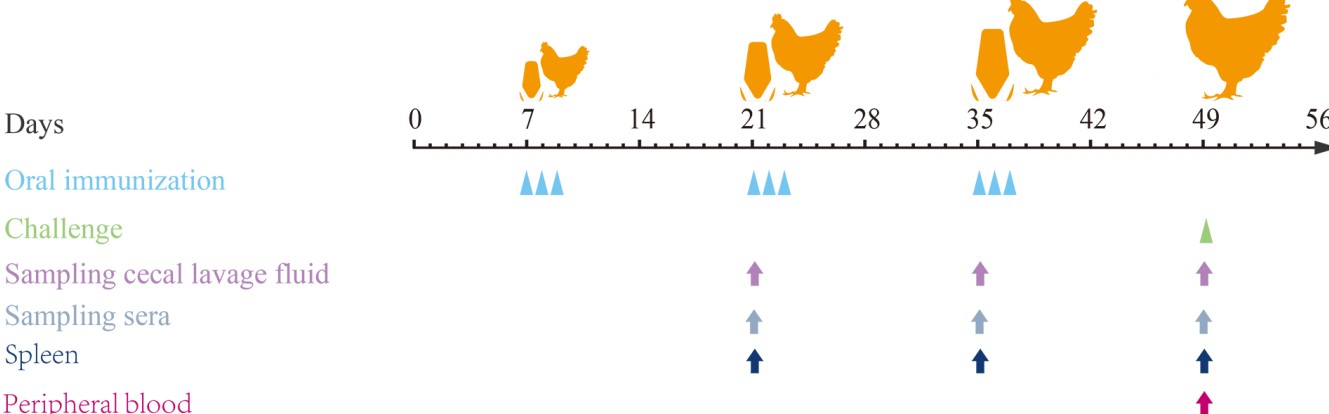

**FIG 11** Schematic procedures for oral immunization, sample collection, and challenge. Oral immunization with recombinant *E. faeaclis* surface-delivering C4D or BC1 domain of EtMIC3 at 21, 22, 23, 35, 36, 37, 49, 50, and 51 days old. Infection experiments were performed at 49 days old. Before secondary and tertiary immunization (at 21 and 35 days old) and before challenging (at 49 days old), sera and cecal lavage fluid were prepared for detection of IgG and sIgA, respectively, and spleen tissues of chickens from each group were sampled for testing proliferation of PBLs. Euthanasia and autopsy were performed at 3 dpi to observe gross pathological lesions.

orally challenged with $4.0 \times 10^4$ *E. tenella*-sporulated oocysts. Animal experiments were performed according to the rules of the Animal Care and Ethics Committees of Northeast Agricultural University, China.

## Antibody responses

At 14 days after each immunization, chickens ($n = 3$) in each group were randomly selected for sampling of peripheral blood and cecal tissues that were used for the preparation of sera and cecal lavage fluids, respectively. The levels of IgG in sera and secreted IgA (sIgA) in the cecal lavage fluid of chickens in each group were detected by ELISA as previously described (20). Briefly, each well in a 96-well plate was coated with 100 µL of C4D or BC1 protein (10 µg/mL) overnight at 4°C. After washing with PBST (0.05% Tween-20 in PBS, pH 7.2), the plate was blocked with 5% skim milk in PBST. Sera (1:300) or cecal lavage fluids (1:200) were added to each well and incubated for 2 h at 37°C. After washing, the plate was reacted with HRP-conjugated goat anti-chicken IgG (1:2,000, Sigma-Aldrich) or IgA (1:5,000, Sigma-Aldrich) for 2 h at 37°C. After washing, 100 µL of o-phenylenediamine (1 mg/mL) and $H_2O_2$ (0.01%) were added to each well, and the reaction was stopped by 2 M $H_2SO_4$. The absorbance was measured at 490 nm using a microplate reader (Bio-Rad, USA).

## mRNA expression levels of cytokines in the spleen

At 14 days after each immunization, chickens ($n = 3$) in each group were randomly selected for sampling spleen tissues. The mRNA expression levels of chicken interferon-γ, chicken interleukin 2, chicken IL-4, chicken IL-10, chicken IL-6, and chicken IL-17 were determined by real-time quantitative reverse transcription PCR (qRT-PCR). Briefly, Trizol (Invitrogen, California, USA) was used to extract total RNA from spleen tissues. Then, cDNA was synthesized from extracted total RNA using the PrimeScript RT reagent kit (TaKaRa Biotech Corp., Dalian, China) according to the manufacturer's instructions. The SYBR Premix Ex Taq II kit (TaKaRa Biotech Corp., Dalian, China) was used for qRT-PCR analysis. The sequences of primer pairs used in this study are listed in Table 3. Each sample was performed in triplicate, and the data were analyzed using the $2^{-\Delta\Delta Ct}$ method.

## Proliferation assay of peripheral blood lymphocytes

Two weeks after the third immunization, PBLs from chickens ($n = 3$) in each group were isolated by using lymphocyte separation medium (Solarbio, Beijing, China) according to the manufacturer's instructions. The prepared lymphocytes were washed three times with RPMI 1640 medium and then cultured in medium supplemented with 10% fetal bovine serum (Gibco, USA), penicillin (100 U/mL), and streptomycin (0.1 mg/mL). Cell viability was estimated by trypan blue dye exclusion. The proliferation of PBLs was determined with the Cell Counting Kit-8 solution (CCK-8) (Bimake, USA). Briefly, cells were added to a 96-well plate ($1.0 \times 10^6$ per well) and incubated for 24 h at 37°C in a 5% $CO_2$ incubator. Cells in each well were stimulated with 10 µg/mL of purified recombinant BC1 or C4D protein for 48 h, and 10 µL of CCK-8 solution was added into each well to incubate for 4 h at 37°C. The value of OD490 nm in each well was detected. All samples were analyzed in triplicate. Cells stimulated with 20 µg/mL of concanavalin A (ConA, Sigma) were used as controls. Cells incubated with culture medium and GST protein were used as negative controls.

## Expression levels of inflammatory factors in the ceca by ELISA

At 7 days post-challenge, the cecal tissues of chickens ($n = 3$) in each group were sampled for quantification of inflammatory cytokines, including IL-1β, IL-6, IL-8, and TNF-α. The concentration of inflammatory cytokines was determined using an ELISA kit (Mlbio, Shanghai, China) as described by the manufacturer's protocol.

## Evaluation of protective efficacy

On the day of the challenge and at 7 days post-challenge, chickens ($n = 8$) per group were randomly weighed to calculate body weight gain, and the percentage increase in weight gain was determined as follows: the percentage increase in weight gain = the average weight gain in the infected group / the average weight gain in the uninfected control group × 100%. At 1 week post-challenge, five chickens per group were randomly selected for evaluating cecal lesion scores, as reported by Johnson and Reid (37). From 7 to 11 days post-infection, 10 chickens were used for oocyst counting by using McMaster's counting technique under a microscope as previously described (38), and the oocyst reduction ratio was calculated using the following formula: oocyst reduction ratio = (the number of oocysts shed by chickens in the challenge control group − the number of oocysts in each immunized group) / challenge control group × 100%. At 7 days post-infection, the cecal tissues were fixed in 10% formalin, embedded with paraffin, sectioned, stained with hematoxylin and eosin, and visualized with a microscope (Olympus, Japan).

## ACKNOWLEDGMENTS

We gratefully acknowledge all the other members working in the Molecular Pathological Lab for their kind coordination and support in experiments.

This study is funded by grants from the National Natural Science Foundation of China (31973003) and the Heilongjiang Natural Science Foundation (LH2022C035).

## AUTHOR AFFILIATIONS

[1]College of Veterinary Medicine, Northeast Agricultural University, Harbin, China
[2]College of Food Science, Northeast Agricultural University, Harbin, China
[3]Heilongjiang Provincial Key Laboratory of Pathogenic Mechanism for Animal Disease and Comparative Medicine, College of Veterinary Medicine, Northeast Agricultural University, Harbin, China

## AUTHOR ORCIDs

Chunli Ma 🔟 http://orcid.org/0000-0002-3830-4400
Dexing Ma 🔟 http://orcid.org/0000-0003-0813-6458

## FUNDING

| Funder | Grant(s) | Author(s) |
|---|---|---|
| MOST \| National Natural Science Foundation of China (NSFC) | 31973003 | Dexing Ma |

## AUTHOR CONTRIBUTIONS

Xinghui Pan, Project administration, Writing – original draft | Rui Kong, Methodology, Software | Qiuju Liu, Data curation, Investigation | Zhipeng Jia, Formal analysis | Bingrong Bai, Investigation | Hang Chen, Data curation | Wenjing Zhi, Methodology | Biao Wang, Software | Chunli Ma, Supervision, Writing – original draft, Writing – review and editing | Dexing Ma, Funding acquisition, Supervision, Writing – original draft, Writing – review and editing

## ETHICS APPROVAL

All animal experiments were conducted at the Veterinary Hospital of Northeast Agricultural University, China. Experimental protocols were carried out according to Ethics Committee for Animal Sciences regulations at Northeast Agricultural University, Heilongjiang Province, China (NEAUEC20210332).

## ADDITIONAL FILES

The following material is available online.

### Open Peer Review

**PEER REVIEW HISTORY (review-history.pdf).** An accounting of the reviewer comments and feedback.

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
