## [Reviewer comments · Microbiology Spectrum]

Microbiology Spectrum

Probiotics *Enterococcus faecalis* surface-delivering key domain of EtMIC3 proteins: Immunoprotective efficacies against *E. tenella* infection in chickens

Xinghui Pan, Rui Kong, Qiuju Liu, Zhipeng Jia, Hang Chen, Wenjing Zhi, Bingrong Bai, Biao Wang, Chunli Ma, and Dexing Ma

Corresponding Author(s): Dexing Ma, Northeast Agricultural University

Review Timeline:

Submission Date:	June 17, 2023
Editorial Decision:	July 23, 2023
Revision Received:	September 11, 2023
Accepted:	September 15, 2023

Editor: Mostafa Zamanian

Reviewer(s): The reviewers have opted to remain anonymous.

Transaction Report:

DOI: <https://doi.org/10.1128/spectrum.02455-23>

July 23, 2023

Prof. Dexing Ma
Northeast Agricultural University
NO600 changjiang Road, Xiangfang Dis
Harbin
China

Re: Spectrum02455-23 (**Probiotics *Enterococcus faecalis* surface-delivering key domain of EtMIC3-proteins: Immunoprotective efficacies against *E. tenella* infection in chickens**)

Dear Prof. Dexing Ma:

Thank you for submitting your manuscript to Microbiology Spectrum. Your manuscript has been evaluated by two reviewers, who each express reasonable (major and minor) suggestions to improve the readability and interpretation of your work. When submitting the revised version of your paper, please provide (1) point-by-point responses to the issues raised by the reviewers as file type "Response to Reviewers," not in your cover letter, and (2) a PDF file that indicates the changes from the original submission (by highlighting or underlining the changes) as file type "Marked Up Manuscript - For Review Only". Please use this link to submit your revised manuscript - we strongly recommend that you submit your paper within the next 60 days or reach out to me. Detailed instructions on submitting your revised paper are below.

Link Not Available

Sincerely,

Mostafa Zamanian

Journals Department
Reviewer comments:

Reviewer #1 (Comments for the Author):

This study by Pan et al investigates the protective effects of the C4D and BC1 domains fused to the dendritic targeting peptide using *E. faecalis* MDXEF-1 strain as a vector against the chicken parasite *E. tenella*. The authors demonstrate the fused proteins are significantly better compared to C4d and BC1 proteins alone at protecting the chicken from the parasite by stimulating the cellular and humoral responses. No significant weight loss and pathologies were observed in the vaccinated birds compared to the vector control treated birds.

Major concerns

1. The relationship between the EtMIC3 protein and the C4D and BC1 domains are not clearly conveyed in the introduction. A

few sentences describing their relationship to the EtMIC3 would be helpful to the readers.

2. Rather than including it in the introduction as unpublished data, the use of the *E. faecalis* strain MDXEF-1 could be included in this manuscript as data since it is the vector or delivery system used for the expression of these fusion proteins. This could be included in the supplementary data.
3. In the results section, the purpose of including the expression data of C4D in *E. coli* BL21 is not clearly articulated in the text. Please include a few sentences to explain the purpose of these experiments.
4. The description for figure 1B is missing in the figure legend.
5. Please specify what lanes 1, 2 and 3 represent in each western blot for figure 2A, B, C and D. Are these replicates of the samples and controls?
6. Indicate if samples E and H in figure 2 are controls in the figure legend.
7. The mRNA transcript levels may not reflect the protein levels, it would be better served if the cytokine levels in the spleen could be measured using an ELISA rather than qRT-PCR.
8. Infiltration of immune cells, destroyed villi and damage to the lamina propria could be indicated by arrows and asterisks in the histology images included in figure 9.
9. Please denote the primer sequences that are underlined indicating restriction enzyme recognition sites in table 3.
10. Were all constructs sequenced to confirm no mutations were incorporated during the amplification of DNA.
11. No experiments were conducted to investigate the additive effects of the two fusion or unfused proteins. Does the administration of both versions of the proteins together provide a more robust response?
12. There are several grammatical and spelling errors throughout the text. Please revise and make the necessary corrections.

Reviewer #2 (Comments for the Author):

1- Can the)BC1(protein fuse with)DCpep(to effectively eliminate the parasite?

2- Can biological treatments replace chemical drugs in the elimination of parasites, as suggested by research and experimental results?

The research focuses on the pressing issue of heavily infected chicken birds caused by this parasite, leading to significant health and economic consequences as well as environmental pollution. I express my gratitude to the researchers for their efforts in investigating this critical area.

Staff Comments:

Preparing Revision Guidelines

Please return the manuscript within 60 days; if you cannot complete the modification within this time period, please contact me. If you do not wish to modify the manuscript and prefer to submit it to another journal, please notify me of your decision immediately so that the manuscript may be formally withdrawn from consideration by Microbiology Spectrum.

Corresponding authors may join or renew ASM membership to obtain discounts on publication fees. Need to upgrade your

membership level? Please contact Customer Service at Service@asmusa.org.

From my perspective, in this research paper:

- 1- Can the (BC1) protein fuse with (DCpep) to effectively eliminate the parasite?
- 2- Can biological treatments replace chemical drugs in the elimination of parasites, as suggested by research and experimental results?

The research focuses on the pressing issue of heavily infected chicken birds caused by this parasite, leading to significant health and economic consequences as well as environmental pollution. I express my gratitude to the researchers for their efforts in investigating this critical area.

Dear Editor Mostafa Zamanian,

How are you!

We are very appreciated for the reviewers' comments. The revisions have already been done in our manuscript as suggested by the reviewers. The revised words were remarked into red color.

Herein I answer the questions as follows:

Reviewer #1 (Comments for the Author):

This study by Pan et al investigates the protective effects of the C4D and BC1 domains fused to the dendritic targeting peptide using *E. faecalis* MDXEF-1 strain as a vector against the chicken parasite *E. tenella*. The authors demonstrate the fused proteins are significantly better compared to C4d and BC1 proteins alone at protecting the chicken from the parasite by stimulating the cellular and humoral responses. No significant weight loss and pathologies were observed in the vaccinated birds compared to the vector control treated birds.

Major concerns

1. The relationship between the EtMIC3 protein and the C4D and BC1 domains are not clearly conveyed in the introduction. A few sentences describing their relationship to the EtMIC3 would be helpful to the readers.

Agree with the reviewer, and several sentences (lines 65-68) were added to describe the relationship between EtMIC3 and C4D and BC1 domains in the revised manuscript.

2. Rather than including it in the introduction as unpublished data, the use of the *E. faecalis* strain MDXEF-1 could be included in this manuscript as data since it is the vector or delivery system used for the expression of these fusion proteins. This could be included in the supplementary data.

Thanks a lot for the reviewer's suggestion, the related sentence was rewritten (lines 78-79) in the

revised manuscript. The data about the isolation and characteristics of *E. faecalis* strain MDXEF-1 was arranged in another article.

3. In the results section, the purpose of including the expression data of C4D in *E. coli* BL21 is not clearly articulated in the text. Please include a few sentences to explain the purpose of these experiments.

The sentence in lines 93-95 was added to explain the purpose of C4D expression in *E. coli* BL21.

4. The description for figure 1B is missing in the figure legend.

The description for figure 1B was added in the revised figure legend.

5. Please specify what lanes 1, 2 and 3 represent in each western blot for figure 2A, B, C and D.

Are these replicates of the samples and controls?

The description for figure 2 was rewritten, and all the lanes were specified.

6. Indicate if samples E and H in figure 2 are controls in the figure legend.

Samples E and H in figure 2 are negative control, which was added in the figure legends.

7. The mRNA transcript levels may not reflect the protein levels, it would be better served if the cytokine levels in the spleen could be measured using an ELISA rather than qRT-PCR.

Thanks a lot for the reviewer's constructive suggestion, and in our subsequent research, cytokine levels in immune organs would be quantified by using ELISA kit.

8. Infiltration of immune cells, destroyed villi and damage to the lamina propria could be indicated by arrows and asterisks in the histology images included in figure 9.

Thanks a lot for the reviewer's suggestion, and several arrows were added in figure 9.

9. Please denote the primer sequences that are underlined indicating restriction enzyme recognition sites in table 3.

Agree with the reviewer, and the note for table 3 was added, showing that underlined sequence represent restriction enzyme recognition sequence.

10. Were all constructs sequenced to confirm no mutations were incorporated during the amplification of DNA.

Yes, all the constructs were sequenced to confirm the correction of target sequence.

11. No experiments were conducted to investigate the additive effects of the two fusion or unfused proteins. Does the administration of both versions of the proteins together provide a more robust response?

Thanks for the reviewer's suggestion. The aim of the present study was to investigate whether BC1 or C4D domain delivered by *E. faecalis* provided protective effects. In the subsequent research work, the combined immunization with two or more kinds of recombinant *E. faecalis* will be further explored.

12. There are several grammatical and spelling errors throughout the text. Please revise and make the necessary corrections.

The manuscript was thoroughly checked, and the grammatical and spelling errors were corrected.

Reviewer #2 (Comments for the Author):

1- Can the BC1 protein fuse with DCpep to effectively eliminate the parasite?

Thanks for the reviewer's question. According to the results from this study, BC1 protein fused with DCpep could partially inhibit *E. tenella* invasion into host cells, and provided protective effects on relieving clinical symptoms of coccidia infection. The BC1 protein fuse with DCpep can not completely eliminate *Eimeria* parasites infection.

2- Can biological treatments replace chemical drugs in the elimination of parasites, as suggested by research and experimental results?

In the past decades, researcher in the field of *Eimeria* research carried on a lot of work on developing novel safe, more effective and easily used biological products to replace traditional chemical drugs. Although no novel products are available commercially till now, we still have confidence to go on exploring more strategies to control *Eimeria* infection.

The research focuses on the pressing issue of heavily infected chicken birds caused by this parasite, leading to significant health and economic consequences as well as environmental pollution. I express my gratitude to the researchers for their efforts in investigating this critical area.

Thanks for the reviewer's encouragement, our group will go on exploring the novel measures to control avian coccidiosis to contribute to a healthier and more sustainable poultry farming industry.

September 15, 2023

Prof. Dexing Ma
Northeast Agricultural University
NO600 changjiang Road, Xiangfang Dis
Harbin
China

Re: Spectrum02455-23R1 (**Probiotics Enterococcus faecalis surface-delivering key domain of EtMIC3-proteins: Immunoprotective efficacies against E. tenella infection in chickens**)

Dear Prof. Dexing Ma:

Your manuscript has been accepted, and I am forwarding it to the ASM Journals Department for publication. You will be notified when your proofs are ready to be viewed.

Sincerely,

Mostafa Zamanian
Editor, Microbiology Spectrum
